# New Records of Fish Parasitic Isopods (Crustacea: Isopoda) from the Gulf of Thailand

**DOI:** 10.3390/ani10122298

**Published:** 2020-12-04

**Authors:** Watchariya Purivirojkul, Apiruedee Songsuk

**Affiliations:** Animal Systematics and Ecology Speciality Research Unit, Department of Zoology, Faculty of Science, Kasetsart University, Bangkok 10900, Thailand; Apiruedee.s@outlook.com

**Keywords:** isopod, *Argathona*, *Cymothoa*, *Smenispa*, *Nerocila*, *Norileca*, Gulf of Thailand

## Abstract

**Simple Summary:**

Parasitic isopods were reported found from marine fishes from many habitat in the world. In Thailand, there is not much study on this parasitic group. This work has compiled all published parasitic isopods documents in Thailand from year 1950 to present include collecting samples from the Gulf of Thailand during the period 2006–2019. New host records were found from four species of parasitic isopods (*Cymothoa eremita*, *Smenispa irregularis*, *Nerocila sundaica*, *Norileca triangulata*) and two species of parasitic isopods (*Argathona macronema*, *Norileca triangulata*) were found first time in the central Indo-Pacific region.

**Abstract:**

From a total of 4140 marine fishes examined, eight species of parasitic isopods were reported from marine fishes in the Gulf of Thailand. These isopods were identified in two families, Corallanidae (*Argathona macronema* and *Argathona rhinoceros*) and Cymothoidae (*Cymothoa eremita*, *Cymothoa elegans*, *Smenispa irregularis*, *Nerocila sundaica*, *Norileca indica* and *Norileca triangulata*). Most of these parasitic isopods were found in the buccal cavity of their fish hosts with one host recorded as follows: *C. eremita* was found from *Nemipterus hexodon*, *C. elegans* was found from *Scatophagus argus*, *N. sundaica* was found from *Saurida tumbil*. The majority of the isopod specimens recorded in this study was *S. irregularis*, which was found in the buccal cavities of five host fish, *Pampus argentius*, *Alepes melanoptera*, *Caranx hippos*, *Parastromateus niger* and *Terapon jarbua*, with a prevalence of 11.67%, 10.43%, 9.78%, 6.10% and 4.21%, respectively. *Argathona rhinoceros* was found in the nasal cavity and branchial cavity of *Epinephelus coioides*, whereas *A. macronema* and *N. triangulata* were found on the skin of *Epinephelus coioides* and *Seriolina nigrofasciata*, respectively. The highest species diversity was found in *E. coioides*, which harbored two species of parasitic isopods, *A. macronema* and *A. rhinoceros*. *Cymothoa eremita*, *C. elegans*, *S. irregularis* and *N. triangulata* were recorded for the first time in the Gulf of Thailand. The reported discovery of *C. eremita*, *S. irregularis*, *N. sundaica* and *N. triangulata* in their fish hosts were new recorded hosts. Moreover, *A. macronema* and *N. triangulata* were found for the first time in the central Indo-Pacific region.

## 1. Introduction

The order Isopoda belongs to the subphylum Crustacea, phylum Arthropoda, with more than 10,300 species found in the deepest oceans to the montane terrestrial habitats [1]. The order Isopoda comprises free-living forms that inhabit various habitats and parasitize mostly fish. Parasitic isopods are typically marine, and usually inhabit the warmer seas [2]. Isopods tend to be small, from 0.5 to 3.0 cm in length, and have a characteristic dorsoventrally flattened body, without a carapace [3]. Most parasitic isopods are ectoparasites, while, for example, Cryptoniscoidea was reported as an endoparasite of crustacean hosts [4]. The large groups of parasitic isopods in marine fishes are members of the suborder Cymothoida, superfamily Cymothooidea. Some families of this group are reported as parasites of fish, both immature forms and adults, although most of them are free-living, viz. Aegidae, Corallanidae, Cymothoidae and Gnathiidae. These parasitic isopods attach to the body surface, in the mouth or on the gills, and are sometimes found in the nasal cavities. Aegidae are distinguished from Cymothoidae by having less modified pereopods. Corallanidae are confined to tropical and subtropical regions [3]. A few corallanid isopods are parasites of fish, such as *Argathona macronema*, which is common in the nasal passages of serranids and lutjanids on the Great Barrier Reef [2]. Cymothoids harm the fish in several ways; mancae (larva stage) feed vocaciously and easily kill fry and fingerlings through the tissue damage they cause [2]. Permanently attached adult isopods can stunt the growth of fish and inhibit reproduction. Those in the gill chamber are usually associated with stunted gills, partly from pressure atrophy and partly from the damage associated with feeding and attachment. They have also been frequently associated with anemia. Those in the mouth affect the development of oral structures and may completely replace the tongue, as with *Ceratothoa oestroides* [2]. Ghani [5] stated that isopod parasites suck the blood of their fish hosts, and the fish becomes weak due to a lack of oxygen and nutrients. A weak fish is more vulnerable to various fatal diseases. Moreover, skin lesions caused by isopods are exposed to secondary bacterial infections.

Their damage to the fish culture population has been reported continuously and seems to be interesting for the study of diversity in different countries. Isopods are associated with many species of commercially important fish around the world and cause significant economic losses [6,7,8,9,10]. In culture systems, the isopod *Alitropus typus* causes high mortality such as in tilapia cage culture in Thailand, with a mortality rate of around 50–100% within 2–7 days after initial infestation [11]. The parasitic isopod group continues to be interesting for scientists in many countries, both regarding wild and culture fish populations, for taxonomic knowledge and economic application. Most reports were in Australia [12,13,14], and appeared in many countries, such as in China, Colombia, India, Indonesia, New Zealand, the northern Arabian Sea, Pakistan, the Philippines and Yemen [5,12,13,14,15,16,17,18,19,20,21,22,23,24,25,26,27].

There have some reports about parasitic isopods in Thailand; the first report regarding a parasitic isopod was recorded in 1950 by Suvatti [28], and *Nerocila phaeopleura*, *Cirolana elongata* (also known as as *Dolicholana elongata*), *Cirolana willeyi*, *Eurydice orientalis* were reported in the Gulf of Thailand. In 1967, Pillai [19] reported the parasitic isopods *Cirolana fluviatilis* and *Alitropus typus* from Thailand. In 1968, Wichiansanka [29] studied fish lice in Songkhla lake and reported that they found three species of isopod—*Rocinela belliceps*, *Cirolana harfordi* and *Excirolana chiltoni*. In 1975, Wichiansanka [30] also studied the biology and distribution of fish lice in Songkhla lake and reported that they found six species of isopod—*Rocinela* sp., *Alitropus* sp., *Cirolana harfordi*, *Excirolana chiltoni*, *Nerocila pigmentata* and *Livoneca vulgaris* (was synonymized with *Elthusa vulgaris*). Previous parasitic isopod reports did not specify the fish host species which attach to fishes’ bodies. However, reports of parasitic isopods with their fish host species were recorded after 1986 by Williams and Williams [31], Bruce and Harrison-Nelson [32], Sirikanchana [33] and Sirikanchana [34] (Table 1).

Regarding non-parasitic species of isopods, *Cirolana fluviatilis*, *C. longistylis*, *Cilicaeopsis whiteleggei*, *Dynamenella yomsii*, *Paradella tomleklek* and *Sphaeromopsis sei* were described in Phuket, Thailand [35,36,37].

This work focuses on surveying the parasitic isopods from the Gulf of Thailand in three areas (upper, central and lower parts) after not having had any reports regarding parasitic isopods on fishes for more than ten years. Although recently, *Nerocila depressa* was found to attach to the bodies of *Sardinella albella* from the estuary area in Thailand [38] and was reportedly found on *Selar crumenophthalmus* [39], that report only surveyed one small area which had an outbreak of isopods in the coastal area. The knowledge from this survey will be applied as taxonomic reference data and prophylaxis in disease management for marine fish culture in this region.

## 2. Materials and Methods

A total of 4140 marine fish specimens were collected from commercial catches (by trawl nets) and individual fishermen (by fishing rod and net) in Chonburi province (13°20′18.45″, 100°55′16.18″ and 12°54′47.80″, 100°47′20.28″) (upper Gulf of Thailand), Prachuap Khiri Khan province (11°7′43.01″, 99°29′11.23″) (central gulf of Thailand) and Surat Thani province (9°20′16.04″, 99°40′52.74″) (lower Gulf of Thailand) (Figure 1) during the period 2006–2019. All fish specimens were dead and were immediately transported in a cool box to the laboratory. Host nomenclature and fish taxonomy are according to FishBase [40]. The skin, nasal cavities, mouth and branchial cavities of each fish were examined. Isopods were removed and keep in 70% ethanol. Mouthparts and appendages of isopods were dissected for identification using the key of [13,15,25,32,41,42,43,44,45,46,47,48,49]. The taxonomy was updated according to the WoRMS catalogue [50].

Isopod specimens were deposited at the Zoological Museum of Kasetsart University (ZMKU). Prevalence and mean intensity were calculated according to Bush et al. [51].

## 3. Results and Discussion

In total, 4140 marine fish specimens from 92 species (*Acanthopagrus berda, Alepes melanoptera, Allenbatrachus grunniens, Anodontostoma chacunda, Arius arius, Brevitrygon imbricata, Caesio cuning, Carangoides armatus,* Caranx *hippos, Cephalopholis boenak, Cephalopholis formosa, Cephalopholis miniata, Cephalopholis sonnerati, Chiloscyllium punctatum, Crenimugil buchanani, Cynoglossus bilineatus, Drepane punctata, Dussumieria elopsoides, Eleutheronema tetradactylum, Ellochelon vaigiensis, Epinephelus areolatus, Epinephelus coioides, Epinephelus erythrurus, Epinephelus faveatus, Epinephelus quoyanus, Epinephelus tauvina, Euthynnus affinis, Gazza minuta, Gerres filamentosus, Gerres oyena, Hilsa kelee, Hyporhamphus quoyi, Johnius dussumieri, Lagocephalus spadiceus, Lates calcarifer, Leiognathus brevirostris, Lutjanus johnii, Lutjanus russellii, Lutjanus vitta, Monacanthus chinensis, Muraenesox bagio, Nemipterus furcosus, Nemipterus hexodon, Nibea soldado, Ophichthus rutidoderma, Otolithes ruber, Pampus argenteus, Pampus chinensis, Parachaetodon ocellatus, Parastromateus niger, Parupeneus crassilabris, Pelates quadrilineatus, Planiliza subviridis, Platax orbicularis, Platax teira, Platycephalus indicus, Plectorhinchus diagrammus, Plectorhinchus pictus, Plotosus canius, Plotosus lineatus, Pomadasys maculatus, Priacanthus tayenus, Psammoperca waigiensis, Psettodes erumei, Rachycentron canadum, Rastrelliger brachysoma, Rastrelliger kanagurta, Sargocentron rubrum, Saurida micropectoralis, Scatophagus argus, Scolopsis monogramma, Scomberoides lysan, Scomberomorus guttatus, Selar crumenophthalmus, Selaroides leptolepis, Seriolina nigrofasciata, Siganus canaliculatus, Siganus javus, Sillago aeolus, Sillago asiatica, Sillago indica, Sillago ingenuua, Sillago maculata, Sillago sihama, Sphyraena obtusata, Stolephorus insularis, Terapon jarbua, Terapon theraps, Trichiurus lepturus, Trypauchen vagina, Tylosurus crocodilus* and *Zebrias quagga*) from Chonburi province (upper Gulf of Thailand), Prachuap Khiri Khan province (central gulf of Thailand) and Surat Thani province (lower Gulf of Thailand) were investigated for parasitic isopods. Seventy samples from 4140 specimens (1.69%) were infected with adult parasitic isopods. They were identified in one suborder, two families, five genera, and eight species, which were found in 11 species of fish (Figure 2 and Figure 3, Table 2) as follows:

Suborder Cymothoida Family Corallanidae
Argathona macronemaArgathona rhinocerosFamily Cymothoidae
Cymothoa eremitaCymothoa elegansSmenispa irregularisNerocila sundaicaNorileca indicaNorileca triangulate


Most of the parasitic isopods from this survey were usually found in the buccal cavity of the fish host, except for *Argathona rhinoceros*, *Argathona macronema*, *Norileca indica* and *Norileca triangulata*. Only *A. rhinoceros* was found in nasal cavities, but in some specimens, they were attached to the branchial cavities of their fish host. *A. macronema* and *N. triangulata* were found on the skin of their fish host, while *N. indica* was found in the branchial cavities. *Smenispa irregularis* was found in five buccal cavities of five hosts, which can separated in three families: Carangidae (*Caranx hippos*, *Alepes melanoptera*, *Parastromateus niger*), Stromateidae (*Pampus argentius*) and Teraponidae (*Terapon jarbua*). Meanwhile, other isopods were found in a single host. *Epinephelus coioides* harbored two species of isopods, *A. macronema* and *A. rhinoceros*. The prevalence of parasitic isopods was highest in *P. argentius*, followed by *A. melanoptera* and *C. hippos*, with a prevalence of 11.67%, 10.43%, and 9.78%, respectively. All of fish in the high prevalence ranking in this study were infected with *S. irregularis*. 

All fish samples were usually found to have only one member of each isopod species, except for *E. coioides* and *C. hippos*, for which two specimens were found—they had a mean prevalence of 1.10 and 1.11 individuals/fish, respectively (Table 2).

The details of the isopods found in this survey were described as follows.

### 3.1. Family Corallanidae

Corallanidae are a small family which are free-living or parasitic to aquatic animals, especially fish [12,46]. All two identified isopod species from the family Corallanidae found in this survey belong to genus *Argathona. Argathona* can be separated from other collanid genera by differences in the mouthpart morphology [12]. *Argathona* have a maxillule with an exopod terminating in an unguis-like point at the base of which lie one or more recurved hook-like processes, an endopod with a truncate lobe, a mandible with a narrow cutting edge, a large triarticulate palp, and a molar process is present [12]. Twelve species of *Argathona* are found in tropical and subtropical marine habitats throughout the Indo-Western Pacific at a depth range of 8-267 m. [46]. Fish temporarily hosting genera of *Argathona* included *Epinephelus*, *Diagramma*, *Cromileptes*, *Plectopomus*, *Variola* (Serranidae), *Pseudolabras* (Labridae), *Lutjanus* (Lutjanidae), *Tetraodon* (Tetraodontidae) and *Muraena* (Muraenidae). This genus was also found as a parasite on turtle, *Chelonia* [46]. Two species of *Argathona*, *A. macronama* amd *A. rhinoceros*, were found in this survey.

### 3.2. Argathona macronema 

(syn. *Aega macronema*, *Alcirona macronema*, *Argatbona macronema*, *Corallana macronema*) [12].

The specimen of *A. macronema* has a length of 15.5 mm, a width of 5 mm, and was found on the skin of *Epinephelus coioides*, although Bruce [12] reported that they occur commonly in the nasal passages of serranids. This specimen is little bit bigger than the specimens from Australia, which have a length of 10–14.4 mm. 

*A. macronema* has been reported as a parasite in *Epinephelus tauvina, Diagramma cinerascens, Pseudolabras* sp., *Trachichtodes affinis, Cromileptes altivelis, Lutjanus argentimaculatus, Lutjanus monostigma, Plectropoma maculatus, Plectropoma laevis* and *Plectropoma leopardus*, and on the eye of a green turtle in Kenya [46,52,53,54].

Bruce [12] stated that this species is a common, widely spread species. It is distributed throughout the Indo-Western Pacific Ocean, including the Red Sea, Gulf of Aqaba, Gulf of Suez, Java, Celebes, New Guinea, Madagascar, Gilbert Islands, Fiji Islands, Kenya, Australia and New Caledonia [46,53]. Miers [55] reported *Corallana macronema*, which was the synonym of *A. macronema* from the Malaysian region. However, Thailand is a newly recorded geographical location for this parasite, and *Epinephelus coioides* is newly recordedhost.

### 3.3. Argathona rhinoceros

(syn. *Livoneca nasicornis*, *Argatbona rhinoceros*, *Cymothoa rhinoceros*, *Gurida coelata*, *Argathona reidi*, *Alcirona pearsoni*) [12].

*Argathona rhinoceros* is usually found in nasal cavities of *Epinephelus coioides*, and the specimen’s length is 6.5–22.0 mm, a width of 2.5–10.0 mm, except for one sample that was found in a gill chamber (length 22.0 mm, width 10.0 mm). This species may grow to be as large as 26.0 mm, as Bruce [12] reported from Australia. The size of this parasite retreived from nasal cavities varied according to size of the fish host. 

*Argathona rhinoceros* has been reported as a parasite in *Tetraodon leopardus, Epinephelus chlorostigma, Epinephelus tauvina, Variola louti, Epinephelus malabaricus, Epinephelus coioides, Epinephelus fuscoguttatus, Epinephelus cyanopodus* and *Plectropoma leopardus* [46,53,54,55]. It is distributed throughout the Indo-Western Pacific, including Aldabra Island, the Persian Gulf, Zanzibar, Sri Lanka, India, Java, Indonesia, Thailand, Australia, the Palau Islands, Eniwetok Atoll and New Caledonia [46]. Although only two species of *Argathona* were reported in the Gulf of Thailand, in another region, *Argathona* spp. were also reported, such as *A. muraenecae*, which was reported as a parasite of *Argyrops spinifer* and *Epinephelus chlorasitgma* in Pakistan [5].

### 3.4. Family Cymothoidae

Isopods in this family are ectoparasites on marine, freshwater and brackish water fishes. Most of them occur in shallow water in tropical and subtropical areas [47]. The position of attachment on the host (buccal cavity, gill chamber) is usually genus- or species-specific [47]. The marine species of the order Isopoda are classified into 12 suborders, among which the suborder Cymothoida includes 29 families. Among the families of the suborder Cymothoida, superfamily Cymothooidea, the family Cymothoidae includes 43 genera [56]. Four genera were found in this study—*Cymothoa*, *Smenispa*, *Nerocila* and *Norileca*.

The genus *Cymothoa* was found to constitute only six species in the southwestern Indian Ocean. In Australia, it has eleven species, and in the central Indo-Pacific region it has nine species [49]. *Cymothoa* is mainly characterized by the general body shape, which is strongly vaulted, with widely separated antennae, a cephalon deeply immersed in pereonite 1, pereonite 7 extending past pleonite 1, a wide pleotelson, pleonite 1 as wide as other pleonites, and uropod rami which are shorter than the pleotelson [22,49].

In this survey, two species of Cymothoa were found: *C. eremita* from *Nemipterus hexodon* and *C. elegans* from *Scatophagus argus*.

### 3.5. Cymothoa eremita

(syn. *Oniscus oestrum*, *Oniscus eremita*, *Cymothoa leschenaultii*, *Cymothoa leaschenaultii*, *Cymothoa limbata*, *Cymothoa mathoei*, *Cymothoa matthaei*, *Cymothoa mathieui*, *Cymothoa edwardsii*, *Cymothoa edwardsi*, *Cymothoa erimitae*, *Cymothoa cinerea*, *Cymothoa cinerius*, *Cymothoa stromatei*) [49].

*C. eremita* was found in the buccal cavity of *Nemipterus hexodon,* with a female length of 21.0 mm and a width of 10.5 mm. In sampling time, many mancas escape from the brood pouch of the mature female and will find a new host. However, many mancas are still attached to the same host. We found them on the outer side of the operculum and also the anterior part of their host fish. Hadfield et al. [49] explained the character of *C. eremita*, in that they have anterolateral projections which extend to half the length of the cephalon, a truncate anterior margin of the cephalon, a pleon as wide as the pereon, uropods which do not reach the posterior margin of the pleotelson, an ischium on pereopod 7 with a bulbous protrusion and small lateral projections on the posterolateral margins of pereonite 1.

*C. eremita* was first collected from Madras in the mouth of *Coryphaena apus* [57]. Moreover, *C. eremita* was reported as a parasite of *Parastromateus niger*, *Psettodes erumei*, *Liza vaigiensis*, *Peprilus paru*, *Pseudanthias evansi*, *Arothron leopardus*, *Tetrodon* sp., *Hime formosana*, *Hime japonica*, *Pampus argenteus*, *Pampus cinereus*, *Siganus canaliculatus*, *Plectorhinchus nigrus*, *Sphyraena obtusata* [44,45,49,58,59,60]. The geographical distribution of this species is widely distributed in the Indo-Western Pacific region, such as Malaysia, Singapore, the Philippines, Indonesia, Australia, Thailand, Japan, China, Ceylon, the Indian Peninsula from Madras to Bombay, Mauritius, the Seychelles, Zanzibar and the Red Sea [43,44,60,61,62,63,64,65]. Although this species has been already reported in Thailand, *Nemipterus hexodon* is a new host recorded for this parasite.

### 3.6. Cymothoa elegans

*C. elegans* was found in the buccal cavity of *Scatophagus argus*. The females found in this study had a length of 14.0–15.5 mm and a width of 5.5 mm. *C. elegans* is characterized by a sublime body that is more than twice as long as it is wide, with nearly parallel sides and very small eyes. The head is sunken into the chest and is wider than it is long, and has fairly short antennae, which are hidden under the head. The fifth segment is the widest. The seventh thorax is the longest and is very broad. The pleon is short and narrow, and the first segment is not hidden. It includes the telson, which is longer and wider than the leon and has posterior rounded corners. The uropods are short.

*C. elegans* was reportedly found on only two host fishes, *Epinephelus fuscoguttatus* and *Scatophagus argus.* The distribution of this parasite was found in Java, Indonesia and the Texas coast [43,66,67,68]. Therefore, the Gulf of Thailand was a new location for this parasite.

### 3.7. Nerocila sundaica 

(syn. *Emphylia ctenopbora*, *Nerocila* (*Emphylia*) *sundaica*) [12].

*Nerocila sundaica* was found in the buccal cavity of *Saurida tumbil.* Females found in this study had a length of 14.0–15.5 mm and a width of 5.5 mm. *Nerocila* is a large genus of the family Cymothoidae including at least 65 species living attached to the skin or on the fins of fishes [7]. *Nerocila* usually have pleonites 1 and 2 with ventrolateral processes, and uropods extending beyond the posterior of pleotelson [24]. Bowman [69] explained the character of *N. sundaica* that have a narrow head. Basal segments of antennae 1 are inflated, and are close to or in contact with one another medially. Distal segments of pereopods 3, 6 and 7 are armed with strong spines on grasping margins. Pereopods 1, 2, 4 and 5 do not have marginal spines, and the dactyls are strongly developed, with that of pereopod 4 being the largest.

Nerocila sundaica was recorded on *Scatophagus argus, Pseudosciaena polyactis, Sciaena* sp., *Eleutheronema* sp., *Mugil* sp., *Otolithes ruber, Therapon jarbua, Engraulis mystax, Serranus gilberti, Pellona indica, Sardinella fimbriata, Carangoides malabaricus*, and *Ilisha melastoma* [12,15,55,69,70,71]. The geographical distribution of N. sundaica is the Persian Gulf, the Bay of Bengal, the Indian Ocean and the northern part of the South China Sea, the Java Sea, the Malaysian region, China, the south coast of Hong Kong, Singapore, the Red Sea and the northern Indian Ocean [12,15,55,69,70,72,73].

### 3.8. Norileca indica

(syn. *Livoneca indica*, *Livoneca ornata*, *Lironeca indica*) [14].

*Norileca indica* was found in the branchial cavity of *Selar crumenophthalmus*. Females found in this study had a length of 2.4–2.9 mm and a width of 1.8–2.2 mm. The body of the parasite twists to one side. The characteristics of genus *Norileca* include pleonites 1 and 2 without ventrolateral processes, and uropods not extending beyond the posterior of pleotelson [14,24]. Pleonite 5 and pleonite 1 are subequal, while pleonite 5 is narrower than pleonite 1 in *N. triangulate*. Uropods are two-thirds the length of pleotelson [14,24]. 

In Thailand, this parasitic isopod was reported only in Selar crumenophthalmus [38,72]. *N. indica* was reportedly found in *Alepes apercna*, *Carangoides malabaricus*, *Decapterus russelli*, *Herklotichthyes* sp., *Nemipterus randalli*, *Rastrelliger kanagurta*, *Secutor insidiator*, *Selar crumenophthalmus* [5,14,23,39,42,74,75,76,77,78,79,80]. These isopods are distributed in the Australia-New Zealand region, the Bay of Bengal, India, Malaysia, Mozambique, Pakistan, the Philippines and Thailand [5,14,23,39,43,74,75,76,77,78,79,80]. 

### 3.9. Norileca triangulata

(syn. *Livoneca triangulata*) [14].

The special characterisitcs of *N. triangulata* include pleonite 5 being manifestly narrower than pleonite 1, and maxilliped palp article 2 being about as long as article 3 [14]. In our study, we found only one female specimen of *N. triangulata* attached to the skin of *Seriolina nigrofasciata*. This female isopod has a length of 15.0 mm and a width of 6.0 mm. 

*N. triangulata* was reported found in *Rastrelliger kanagurta*, *Sardinella gibbosa*, *Parexocoetus brachypterus*. This species is distributed aorung Tanimdao Island, the Philippines, Australia and the southeast coast of India [5,14,17,24,81]. The Gulf of Thailand is a new geographical record for this parasite and *Epinephelus coioides* is a newly recorded host.

### 3.10. Smenispa irregularis

(syn. *Cymothoa irregularis*, *Cymothoa paradoxa*, *Enispa irregularis*) [14,82].

*S. irregularis* was found in five species of fish, *Caranx hippos*, *Caranx malam*, *Pampus argentius*, *Parastromateus niger* and *Terapon jarbua*, in this study. Most of them were found in the buccal cavity of their fish host. Females found in this study had a length of 16.0–22.5 mm and a width of 8.0–12.0 mm. Martin et al. [82] stated that the diagnostic characteristics of the genus are the strongly vaulted body, a cephalon embedded in pereonite 1, antennula shorter than the antenna, bases set wide apart, the pereon and pleon being co-linear with the sub-parallel lateral margins, pereopods lacking carina on the bases, and the endopods of pleopods 3–5 having large folds. The size of the ovigerous female in our study seems to be smaller when compared with Martin et al. [82] (20–24 mm). 

Previously, *E. irregularis* had only two host records, *Psettodes erumei* and *Caranx carangus* [14]. Recently, Martin et al. [82] changed the nomenclatural from *Enispa irregularis* to *Smenispa irregularis*, and reported the host as *Acanthopagrus latus* (Sparidae) from Western Australia. However, Miers [55] stated that *Cymothoa irregularis* is common on fishes in the seas of Amboina, Malaysia. Moreover, Martin et al. [82] noted that *Smenispa irregularis* is known to occur on host species from the families Sparidae, Carangidae, and Psettodidae, suggesting the species has low host specificity. *E. irregularis*’s distribution included Ambon Island, Arafura Sea, Atlantic coast of Panama, Carnarvon, Jakarta Bay, Northern Territory, Singapore, Thailand and Western Australia [14,42,70,82,83,84].

*Caranx hippos, Alepes melanoptera, Pampus argentius, Parastromateus niger* and *Terapon jarbua* were new host records of *E. irregularis.*

In this survey, we did not find *Livoneca* sp., although this genus has been recorded in Thailand for a long time. Smit et al. [85] stated that of the approximately 60 species that had been placed in *Livoneca* up to 1990, most were relocated to *Elthusa* and *Ichthyoxenus*, with only three species now remaining in the genus [14]. From our results, two species of parasitic isopods previously had their genus as *Livoneca* but changed to another genus—*Argathona rhinoceros* (syn. *Livoneca nasicornis*) and *Norileca triangulata* (syn. *Livoneca triangulata*).

The size of the parasitic isopods on their host in this study in some specimens was not in same range as the literature cited, such as *A. macronema* and *S. irregularis*. This can explained by the study by Leonardos and Trilles [86], who studied host–parasite relationships between parasitic isopod *Mothocya epimerica* on sand smelt *Atherina boyeri* and found that parasite size increased with host size. The prevalence of isopod infections from this study was only the guideline for advance research because we randomised the marine fish specimens discontinuously, because the prevalence of parasites varied according to the month, and the sex and size of hosts [87].

Low host specificity for parasitic isopod to their fish hosts was also confirmed. The parasitic isopods in each fish species from the literature cited seem to be different from this study. For example, *Caranx hippos* from the Caribbean area was found to harbour *Cymothoa oestrum* inside the mouth [47]. *Parastromateus niger* was found to harbour *Cymothoa eremita* [57]. *Pampus argenteus* was found to harbour *Cymothoa eremita* [23,49]. For only two hosts, *A. rhinoceros* was found in the nasal cavities of *Epinephelus coioides*, which is the same as the report in New Calidonia [53], and *Scatophagus argus* hosted the parasitic isopod *Cymothoa elegans,* which is the same as the report in Indonesia [67].

The effects of parasitic isopods on their fish hosts have been explained in many records. However, in most cases, the mean intensity is usually only one isopod per infected fish. In this case, infection in low prevalence in a fish population does not seem to affect the economy. However, in some cases, they can change fish behavior, anatomy, or morphology [88]. The swimming capacity of the fish was also found to be affected by parasitic isopods [89]. For instance, *Cymothoa exigua* sucks so much blood from its host fish’s tongue that the tongue atrophies and is destroyed and the isopod remains attached to the remaining tongue stub and is used by the host as a replacement tongue for food manipulation [90]. Although many parasitic isopods feed on fish blood, there are some exceptions, such as *Argathona macronema*, which feeds on fish mucus, not blood [8]. The low mean intensity with low prevalence seems to not be a danger for marine aquaculture, but the isopod population can increase their number rapidly if this habitat has a high population of fish hosts, such as a fish culture system. They affect fish growth and production, causing economic losses.

Lester [2] explained the life cycle of an isopod of a gravid female which releases eggs into a brood pouch. The eggs embryonate, hatch and undergo two or more moults to form manca or pullus II stage. In this study, some isopod specimens of *S. irregularis* were found as many mancae from the brood pouch of a gravid female (Figure 2). The post-mancal juvenile stages (sometimes referred to as the aegathoid stage) have only six pairs of legs, large compound eyes, and highly setose pleopods for active swimming. The juveniles will attach themselves to a convenient fish host and eventually attach to the preferred host-species [46]. After a short free-swimming period, they are parasitic and need to find a fish to take their first meal within one to two days or they will die. From this survey, we found that a specimen of *Nemipterus hexodon* was attached with many mancae (Figure 2A,B), and this fish specimen had the isopod *Cymothoa eremita* in its mouth (Figure 2A,B). This finding seems to correspond with that of Lester [2], as in the tongue biters group such as *Ceratothoa* spp., the mancae move to the preferred site and remain attached to the fish. 

In the cymothoids group, they have a short free-living planktonic phase and then attach to fish hosts. Heavy infestations of parasitic juveniles have the potential to kill small fingerlings. So, in a culture system, prevention may done by eradicate immature forms of isopod when they are planktonic [11].

Other than the data regarding the abundance of parasitic isopods in Thailand as seen in the literature review above, the areas which connect to the Gulf of Thailand that have parasitic isopod reports, such as the Malaysian region [55] and the South China Sea [15], were compared with the results of the parasitic survey in this study to discuss the new geographical record of isopod parasites in this region. 

Smit et al. [85] mapped the distribution of the marine cymothoids using Spalding et al. [91], and they found that the highest diversity resides within the tropical regions of the central Indo-Pacific, with 79 marine Cymothoidae isopods. Comparing this with our data, all parasitic isopods that we found in the survey were already recored in the central Indo-Pacific region, except for *Norileca triangulata*, which was distributed around Tanimdao Island, the Philippines, Australia and southeast coast of India.

## 4. Conclusions

The survey found eight species of parasitic isopod—*Argathona macronema*, *Argathona rhinoceros*, *Cymothoa eremita*, *Cymothoa elegans*, *Nerocila sundaica*, *Norileca indica*, *Norileca triangulata* and *Smenispa irregularis*. Nine new host records of five parasites were also reported, *Argathona macronema* from host *Epinephelus coioides*, *Cymothoa eremita* from host *Nemipterus hexodon*, *Cymothoa elegans* from host *Scatophagus argus*, *Nerocila sundaica* from host *Saurida tumbil*, *Norileca triangulata* from host *Seriolina nigrofasciata* and *Smenispa irregularis* from *Alepes melanoptera*, *Caranx hippos*, *Pampus argentius*, *Parastromateus niger* and *Terapon jarbua*. However, in this study, the praniza stage of isopod *Gnathia* spp. (family Gnathiidae) (Figure 3M) was also found in 15 species of fishes (*Cephalopholis formosa*, *Cephalopholis miniata*, *Cephalopholis sonnerati*, *Epinephelus areolatus*, *Epinephelus coioides*, *Epinephelus erythrurus*, *Epinephelus quoyanus*, *Lutjanus johnii*, *Lutjanus russellii*, *Pelates quadrilineatus*, *Plectorhynchus pictus*, *Scatophagus argus*, *Seriolina nigrofasciata*, *Sillago aeolus* and *Sillago sihama*).

*Argathona macronema*, *Cymothoa eremita*, *Cymothoa elegans* and *Norileca triangulata* were newly recorded in Thailand. According to Spalding et al. [91], *Norileca triangulata* was found for the first time in the central Indo-Pacific region.

## Figures and Tables

**Figure 1 animals-10-02298-f001:**
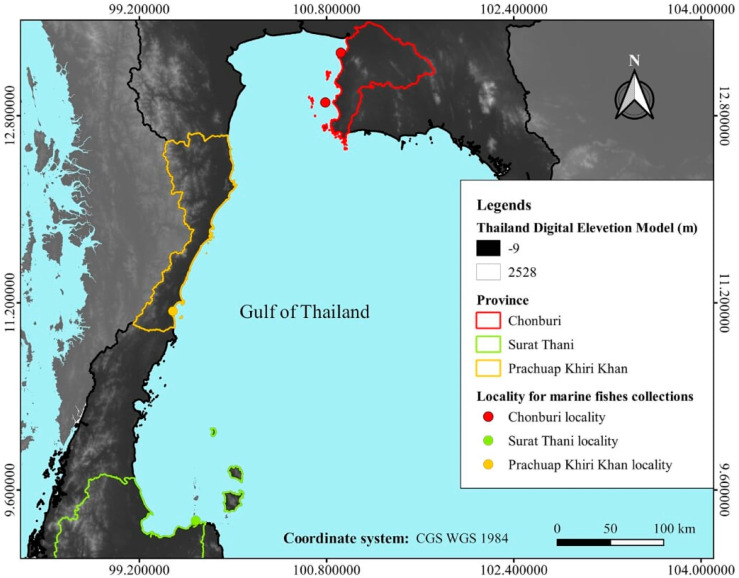
Map of study area in the Gulf of Thailand.

**Figure 2 animals-10-02298-f002:**
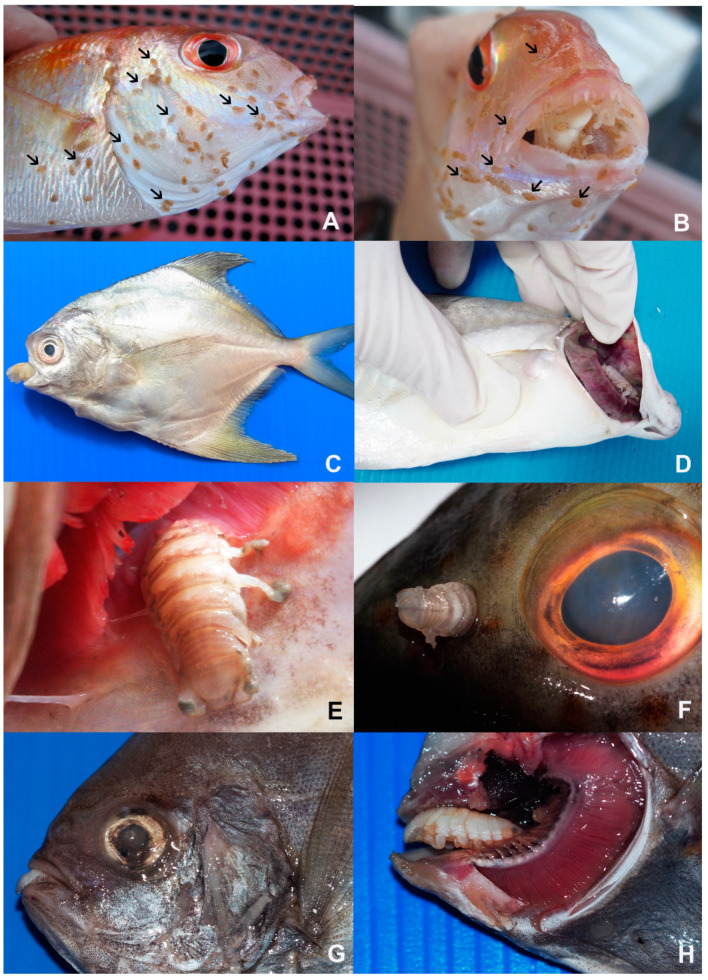
Various species of parasitic isopods attached to their fish host. (**A**,**B**) *Cymothoa eremita* in the buccal cavity of *Nemipterus hexodon*, many mancae (arrows) attached to same fish host. (**C**) *Smenispa irregularis* in the buccal cavity of *Pampus argentius*. (**D**) *Smenispa irregularis* in the gill chamber of *Caranx hippos*. (**E**) *Argathona rhinoceros* in the gill chamber of *Epinephelus coioides*. (**F**) *Argathona rhinoceros* in the nasal cavity of *Epinephelus coioides*. (**G**,**H**) *Smenispa irregularis* in the buccal cavity of *Parastromateus niger*, (**H**) after removal of the operculum).

**Figure 3 animals-10-02298-f003:**
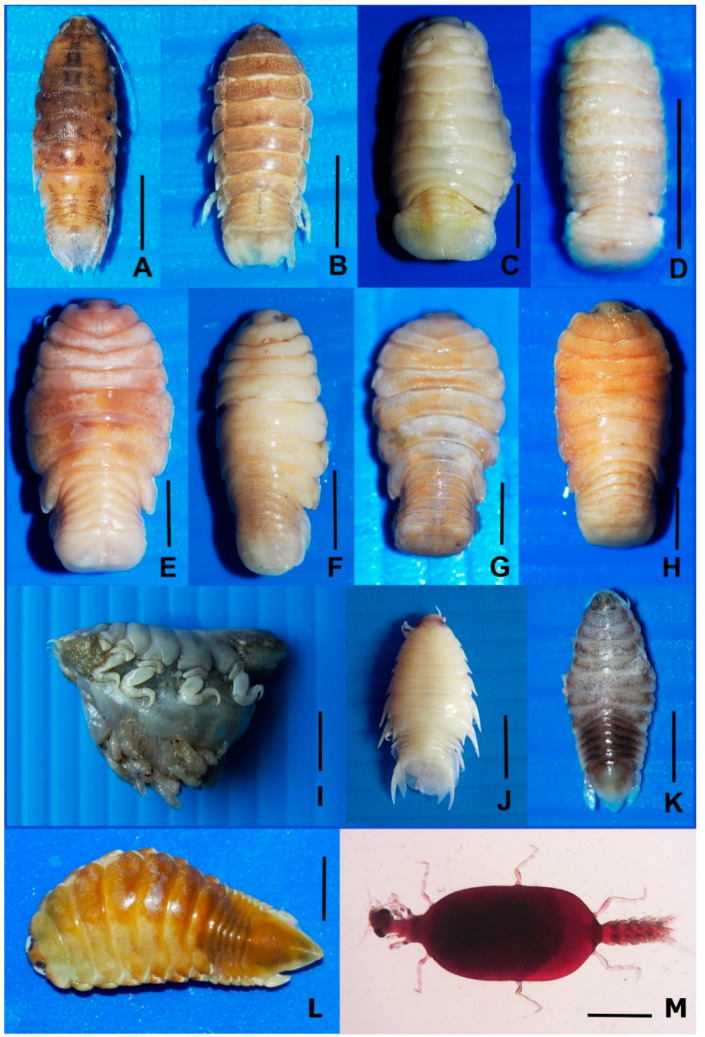
Parasitic isopods found in this study (**A**–**L**). scale bar = 5 mm.). (**A**). *Argathona macronema* from *Epinephelus coioides*. (**B**). *Argathona rhinoceros* from *Epinephelus coioides*. (**C**). *Cymothoa eremita* from *Nemipterus hexodon*. (**D**). *Cymothoa elegans* from *Scatophagus argus*. (**E**). *Smenispa irregularis* from *Caranx malam*. (**F**). *Smenispa irregularis* from *Pampus argentius*. (**G**). *Smenispa irregularis* from *Parastromateus niger*. (**H**). *Smenispa irregularis* from *Terapon jarbua*. (**I**). *Smenispa irregularis* from *Caranx malam*, with many mancae in the brood pouch. (**J**). *Nerocila sundaica* from *Saurida tumbil*. (**K**). *Norileca triangulata* from *Seriolina nigrofasciata*. (**L**). *Norileca indica* from *Selar crumenophthalmus*. (**M**). *Gnathia* sp. (praniza stage) from various marine fishes species (scale bar = 0.4 mm).

**Table 1 animals-10-02298-t001:** Parasitic isopods with their host recorded in Thailand from the literature cited.

Parasitic Isopod	Fish Host	Reference
*Anilocra* sp.	*Epinephelus* sp.	[34]
*Cirolana* sp.	*Monacanthus* sp.	[34]
*Codonophilus* sp. (accepted as *Ceratothoa* sp.)	*Stolephorus indicus*	[34]
*Excirolana chiltoni*	*Cheilinus chlorourus*	[33]
	*Kyphosus cinerascens*	[33]
	*Plectropoma maculatum* (accepted as *Plectropomus areolatus*)	[33]
	*Plectorhynchus pictus* (accepted as *Plectorhinchus pictus*)	[33]
	*Scolopsis monogramma*	[33]
	*Siganus oramin* (accepted as *Siganus canaliculatus*)	[33]
	*Plectropoma maculatum* (accepted as *Plectropomus maculatus*)	[34]
	*Scolopsis monogramma*	[34]
*Glossogobius* sp.	*Parexocoetus mento*	[31]
*Gnathia* sp. (praniza stage)	*Cephalopholis boenak*	[33]
	*Lutianus vaigiensis* (accepted as *Lutjanus fulvus*)	[33]
	*Scolopsis ciliatus* (accepted as *Scolopsis ciliata*)	[33]
	*Scolopsis monogramma*	[33]
	*Stethojulis phekadopleura* (accepted as *Stethojulis trilineata*)	[33]
	*Halichoeres nigrescens*	[33]
	*Abudefduf saxatilis*	[33]
	*Drepane punctata*	[34]
*Livoneca circularis* (accepted as *Ryukyua circularis*)	*Amblygaster sirm*	[31]
*Livoneca vulgaris* (accepted as *Elthusa vulgaris*)	*Mugil dussumieri* (accepted as *Planiliza subviridis)*	[34]
*Mothocya renardi*	*Strongylura leiura*	[31]
*Nerocila depressa*	*Cyclocheilichthys apogon*	[32]
	*Engraulis* sp.	[32]
*Nerocila* *loveni*	*Leiognathus* sp.	[32]
*Nerocila phaiopleura*	*Chirocentrus* sp.	[32]
	*Sardinella fimbriata*	[32]
	*Harengula* sp.	[32]
*Nerocila pigmentata*	*Clupea* spp.	[34]
*Nerocila* *serra*	-	[32]
*Nerocila sundaica*	*Sciaena* sp.	[32]
	*Mugil* sp.	[32]

**Table 2 animals-10-02298-t002:** Parasitic isopod and their fish host with site of infection, prevalence (P) and mean intensity (M.I.) from the Gulf of Thailand (in this study).

Parasitic Isopod	Fish Host	Site of Infection	P (%)	M.I.
*Argathona macronema*	*Epinephelus coioides*	skin	1.48	1.00
*Argathona rhinoceros*	*Epinephelus coioides*	nasal cavities/branchial cavity	7.41	1.10
*Cymothoa eremita*	*Nemipterus hexodon*	buccal cavity	0.26	1.00
*Cymothoa elegans*	*Scatophagus argus*	buccal cavity	3.31	1.00
*Smenispa irregularis*	*Caranx hippos*	buccal cavity/skin	9.78	1.11
	*Alepes melanoptera*	buccal cavity	10.43	1.00
	*Pampus argenteus*	buccal cavity	11.67	1.00
	*Parastromateus niger*	buccal cavity	6.10	1.00
	*Terapon jarbua*	buccal cavity	4.21	1.00
*Nerocila sundaica*	*Saurida micropectoralis*	buccal cavity	3.08	1.00
*Norileca indica*	*Selar crumenophthalmus*	branchial cavity	6.67	1.00
*Norileca triangulata*	*Seriolina nigrofasciata*	skin	1.11	1.00

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
