# Peer review of "New Records of Fish Parasitic Isopods (Crustacea: Isopoda) from the Gulf of Thailand"

_animals, 2020, doi:10.3390/ani10122298_

Round 1
Reviewer 1 Report
The authors evaluated a large number of fish species from the gulf of Thailand for parasitic isopods. This manuscript contains valuable information about the isopod parasites of fish in the Gulf of Thailand. The authors provide elegant images of the parasites found in their survey. The authors have evaluated an impressive number of fish for infection, and their work can provide the basis for further research on these species.
There are some issues with the writing, with sentence fragments throughout. In both the introduction and discussion there are a few paragraphs (indicated below) that would be better presented in table form. I also think that it would be valuable for the data to be provided with this publication, including the date and location (at least region) for the prevalence estimates. This could either be in raw data form, or summarized by region and date.
See specific line edits below:
Line 59-90 – I think these paragraphs might be better represented as a table
Line 99-100 – Could you list the types of fishing practices used by the commercial and individual fisherman that collected the fish? And how were the specimens treated from collection to being processed in your lab?
Line 102-103 – Is it possible to provide a supplemental table of the data on infection prevalence across time and location?
Line 346-348 – are you able to report on the seasonal variance in infection in some of the species you have surveyed?
Line 390-403 – I am not sure that this paragraph is needed.
Author Response
Line 59-90 – I think these paragraphs might be better represented as a table
Response : already add Table 1
Line 99-100 – Could you list the types of fishing practices used by the commercial and individual fisherman that collected the fish? And how were the specimens treated from collection to being processed in your lab?
Response : add “(by trawl nets)” after commercial catches and add “(by fishing rod and net)” after individual fishermen
Add “All fish specimens were dead and were immediately transported in a cool box to the laboratory.”
Line 102-103 – Is it possible to provide a supplemental table of the data on infection prevalence across time and location?
Response : Add supplemental table already
Line 346-348 – are you able to report on the seasonal variance in infection in some of the species you have surveyed?
Response : Actully, in the rainy season, there will be a monsoon, the fishermen could not caught the fishes in the Gulf of Thailand. And in June-September, Department of Fisheries Thailand announces the implementation of resource management measures during the period of egg-laying fish and larvae in the inner Gulf of Thailand. The fisherman can’t caught the fishes in this period. So, we did not collect data separated in seasonal.
Line 390-403 – I am not sure that this paragraph is needed.
Response : delete line 390-403 already
Reviewer 2 Report
Dear Authors,
below my comments about the manuscript ID: animals-1009403
Type of manuscript: Article
Title: Distribution of adult parasitic Isopods (Crustacea: Isopoda) from the Gulf of Thailand.
Authors: Watchariya Purivirojkul , Apiruedee Songsuk Submitted to section: Wildlife.
The work is interesting and beneficial for the researchers working in similar area.
It will certainly be an excellent updated contribution on the knowledge of the distribution of isopods in the marine fish population of Thailand.
It is a huge job, with thousands of fish checked that will certainly be considered also in the future.
Tables and pictures of isopods are clear and effective. I appreciate the effort to assemble numerous bibliographic information, very appropriate and updated.
Important: precisely due to the fact that it is a great work that will probably remain as a reference regarding the prevalence of marine isopods in fish in Thailand, in the materials and methods (very short!) I would give more space on the identification keys used both to determine the species of fish and the species of isopods found. What sources do you use: i.e. Fishbase / FAO key for fish? For isopods which keys? even if you have cited some bibliographic sources, I would be a little more precise. The taxonomy is updated i.e. according to the WoRMS catalogue, others?
Do tu seasonal variation of the prevalence of isopods in fish, cuold be be useful to specify this fact if available regarding the fish sampling.
Furthermore, considering the commercial implications of fish parasitized with isopods, which have depreciated, it would be useful in the long line of fish reported in the results and discussion chapter, to distinguish commercial fish (edible) from those that have no commercial value.
Minor revisions:
Title: “Distribution of adult parasitic Isopods (Crustacea: Isopoda) from the Gulf of Thailand”, could be changed in : “Prevalence of fish parasitic Isopods (Crustacea: Isopoda) from the Gulf of Thailand: an update” or “New record of fish parasitic Isopods (Crustacea: Isopoda) from the Gulf of Thailand”.
Line 9:three families.......Corallanidae and Cymothoidae..probably two!
Line 12-13: as follows: change Calibri in Times new roman character. Use the same conversion for line: 35, 37, 67,68, 76-88, 91-97, 233-236, 249-253, 258-260, 266-271, 288-290, 300, 311-312, 352-354, 366-368, 378-379, 404-405.
Line 28: The order isopoda is a diverse group of subphylum… not very clear. Better change in: The order isopoda belongs to the subphylum…
Line 42: additionally, consider also, as cited in numerous scientific works, that skin lesions caused by isopods are exposed to secondary bacterial infections.
Line 53-54: Alitropus typus cause the high mortality such as in tilapia cage culture in Thailand, the isopod cause the mortality with the rate 50-100% within 2-7 day after initial infestation. Could be correct and simplified as follows: “Alitropus typus caused the high mortality such as in tilapia cage culture in Thailand, with a mortality rate around 50-100% within 2-7 day after initial infestation”.
Line 112: fish specimens from 92 species: could be more useful to quantify how many fish for each species.
Line 136: found from 11 species of fish. But in table n. 1 are reported 12 species, why?
Line 152: figure 2 legenda: …many mancae?
Line 450: 2005 must be written in bold type. Also check the years on the following lines: 453, 465, 468, 490, 507, 510, 514, 517, 522, 526, 538, 542, 560, 589, 618, 621, 626, 679, 686.
Author Response
Important: precisely due to the fact that it is a great work that will probably remain as a reference regarding the prevalence of marine isopods in fish in Thailand, in the materials and methods (very short!) I would give more space on the identification keys used both to determine the species of fish and the species of isopods found. What sources do you use: i.e. Fishbase / FAO key for fish? For isopods which keys? even if you have cited some bibliographic sources, I would be a little more precise. The taxonomy is updated i.e. according to the WoRMS catalogue, others?
Response : add “of isopods” after Mouthparts and appendages
add “The taxonomy is updated according to the WoRMS catalogue [50]”
Minor revisions:
Title: “Distribution of adult parasitic Isopods (Crustacea: Isopoda) from the Gulf of Thailand”, could be changed in : “Prevalence of fish parasitic Isopods (Crustacea: Isopoda) from the Gulf of Thailand: an update” or “New record of fish parasitic Isopods (Crustacea: Isopoda) from the Gulf of Thailand”.
Response : change the title to “New record of fish parasitic Isopods (Crustacea: Isopoda) from the Gulf of Thailand”
Line 9: three families.......Corallanidae and Cymothoidae..probably two!
Response : change to “two families”
Line 12-13: as follows: change Calibri in Times new roman character. Use the same conversion for line: 35, 37, 67,68, 76-88, 91-97, 233-236, 249-253, 258-260, 266-271, 288-290, 300, 311-312, 352-354, 366-368, 378-379, 404-405.
Response : already edited as reviewer suggested
Line 28: The order isopoda is a diverse group of subphylum… not very clear. Better change in: The order isopoda belongs to the subphylum…
Response : change to “The order isopoda belongs to the subphylum…”
Line 42: additionally, consider also, as cited in numerous scientific works, that skin lesions caused by isopods are exposed to secondary bacterial infections.
Response : Add “skin lesions caused by isopods are exposed to secondary bacterial infections”
Line 53-54: Alitropus typus cause the high mortality such as in tilapia cage culture in Thailand, the isopod cause the mortality with the rate 50-100% within 2-7 day after initial infestation. Could be correct and simplified as follows: “Alitropus typus caused the high mortality such as in tilapia cage culture in Thailand, with a mortality rate around 50-100% within 2-7 day after initial infestation”.
Response : change to “Alitropus typus caused the high mortality such as in tilapia cage culture in Thailand, with a mortality rate around 50-100% within 2-7 day after initial infestation”
Line 112: fish specimens from 92 species: could be more useful to quantify how many fish for each species.
Response : Add data in supplementary file
Line 136: found from 11 species of fish. But in table n. 1 are reported 12 species, why?
Response : Epinephelus coioides was found 2 species of isopods (Argathona macronema and Argathona rhinoceros) so number of fish is 11.
Line 152: figure 2 legenda: …many mancae?
Response : add arrow to point at some mancae
Line 450: 2005 must be written in bold type. Also check the years on the following lines: 453, 465, 468, 490, 507, 510, 514, 517, 522, 526, 538, 542, 560, 589, 618, 621, 626, 679, 686.
Response : already edited